# Synthesizing Disparate LiDAR and Satellite Datasets through Deep Learning to Generate Wall-to-Wall Regional Inventories for the Complex, Mixed-Species Forests of the Eastern United States

Elias Ayrey [1,2,*] , Daniel J. Hayes [2] , John B. Kilbride [3] , Shawn Fraver [2] , John A. Kershaw, Jr. [4] , Bruce D. Cook [5] and Aaron R. Weiskittel [6]

[1] Pachama Inc., 1435 48th Ave, San Francisco, CA 94122, USA
[2] School of Forest Resources, University of Maine, 5755 Nutting Hall, Orono, ME 04469, USA; daniel.j.hayes@maine.edu (D.J.H.); shawn.fraver@maine.edu (S.F.)
[3] College of Earth, Ocean, and Atmospheric Sciences, Oregon State University, 114 Wilkinson Hall, Corvallis, OR 97331, USA; kilbridj@oregonstate.edu
[4] Faculty of Forestry and Environmental Management, University of New Brunswick, P.O. Box 4400, 28 Dineen Drive, Fredericton, NB E3B5A3, Canada; kershaw@unb.ca
[5] NASA Goddard Space Flight Center, Biospheric Sciences Laboratory, Code 618, Greenbelt, MD 20771, USA; bruce.cook@nasa.gov
[6] Center for Research on Sustainable Forests, University of Maine, 5755 Nutting Hall, Orono, ME 04469, USA; aaron.weiskittel@maine.edu
* Correspondence: elias.ayrey@maine.edu

**Abstract:** Light detection and ranging (LiDAR) has become a commonly-used tool for generating remotely-sensed forest inventories. However, LiDAR-derived forest inventories have remained uncommon at a regional scale due to varying parameters among LiDAR data acquisitions and the availability of sufficient calibration data. Here, we present a model using a 3-D convolutional neural network (CNN), a form of deep learning capable of scanning a LiDAR point cloud, combined with coincident satellite data (spectral, phenology, and disturbance history). We compared this approach to traditional modeling used for making forest predictions from LiDAR data (height metrics and random forest) and found that the CNN had consistently lower uncertainty. We then applied the CNN to public data over six New England states in the USA, generating maps of 14 forest attributes at a 10 m resolution over 85% of the region. Aboveground biomass estimates produced a root mean square error of 36 Mg ha$^{-1}$ (44%) and were within the 97.5% confidence of independent county-level estimates for 33 of 38 or 86.8% of the counties examined. CNN predictions for stem density and percentage of conifer attributes were moderately successful, while predictions for detailed species groupings were less successful. The approach shows promise for improving the prediction of forest attributes from regional LiDAR data and for combining disparate LiDAR datasets into a common framework for large-scale estimation.

**Keywords:** LiDAR; airborne laser scanning; enhanced forest inventory; aboveground biomass; forest carbon; deep learning; Maine; New Hampshire; Vermont; Massachusetts; Connecticut; Rhode Island

## 1. Introduction

### 1.1. Overview

Over the past two decades, light detection and ranging (LiDAR) has become a common tool for developing spatially-explicit forest inventories [1]. Measurements of point cloud datasets derived from LiDAR can be used to predict useful forest inventory attributes such as biomass, stem volume, tree count, and species [2–4]. These inventories are useful for a wide range of applications, including assessing carbon stocks [5], assisting in precision forestry [6], and predicting wildlife habitat [7,8].

Forest inventories are typically developed using the area-based approach [9], where the forest is segmented into a series of grid cells ranging from 10 m to 1 ha in size (e.g., [10]). First, the LiDAR point cloud and the desired forest attribute (e.g., stem density) are each measured in a sample of grid cells. Next, predictive models, either parametric or non-parametric, are developed relating the field measurements to the LiDAR measurements. Finally, these models can be applied to every grid cell across a landscape to produce wall-to-wall estimates for attributes of interest. The resulting maps are referred to as enhanced forest inventories (EFIs). A major challenge with the area-based approach is that the final EFI is often limited to local predictions due to variation in the underlying ground and LiDAR data specifications (e.g., [11]).

While LiDAR data are becoming increasingly available to the public, few studies have emphasized mapping whole regions (e.g., [12]) while focusing instead on specific municipalities or individual parcels. One example of regional LiDAR modeling occurred in Sweden, which recently developed nation-wide forest inventory maps at a 12.5 m resolution [13], while similar maps have also been generated in Finland [14]. In Canada, large portions of Alberta's forests have had their vegetative functional groups mapped [15], and in New Brunswick, a provincial effort has resulted in near wall-to-wall LiDAR inventories [16]. We note that each of these examples make use of largely homogeneous LiDAR datasets, which minimizes the potential challenges identified below.

### 1.2. The Current Approach

One common difficulty in generating regional LiDAR inventories is that many regions are comprised of a patchwork of LiDAR datasets acquired with various specifications, and with forest analytics often as a secondary objective. This is particularly problematic because the traditional approach for measuring a LiDAR point cloud for the development of an EFI model involves taking a series of summary statistics describing point heights and their vertical distributions. These include measures of the mean, variance, and vertical quantiles, as well as proportions of points that fall above certain height thresholds [17,18]. Unfortunately, these traditional metrics suffer from several drawbacks, such as (1) a high degree of collinearity, (2) a propensity to change among acquisitions based on LiDAR sampling design (e.g., pulse density), (3) a propensity to change based on forest phenology, and (4) limited ecological inference [19].

Several software suites exist for extracting height features from LiDAR, each producing upwards of 50 metrics, including the heights of every 10th percentile [20]. While powerful predictors, many of these metrics are also highly correlated, creating a risk of model overfitting and overspecification without careful model selection [21,22]. Many studies make use of all available predictors and report on those that are most important; however, some modeling techniques are unreliable for ranking highly collinear features. In some cases, this may impact perceived feature importance and not be optimal depending on the use-case.

A standard measure for assessing LiDAR quality is pulse density, which refers to the number of laser pulses landing within a given area (pls m$^{-2}$). Pulse density can vary both between and within LiDAR acquisitions, and frequently, regional LiDAR collections consist of many acquisitions in which the pulse density varies by up to an order of magnitude. Many studies have found that varying pulse density can adversely affect EFI predictions across different LiDAR data sets. In particular, when the pulse density drops below an ecosystem-specific optimal threshold for deriving height metrics, model performance may be degraded. For example, Gobakken and Næsset [23] noted that area-based predictions are strongly affected by pulse density. Hansen et al. [24] determined that EFI estimates could be subject to bias if predictions were made on point clouds with densities different than those used to train the model, particularly those with lower densities. Other differences in acquisition parameters related to LiDAR sampling design—such as sensor type, pulse frequency, and flight altitude—can also result in different height features being generated over the same area of forest [25,26].

Seasonality also has a major impact on LiDAR height features [22,27,28], particularly in deciduous forests where the presence of leaves can result in LiDAR beams being intercepted higher in the canopy [29]. In the United States, there is no commonly accepted federal standard for acquiring data during leaf-off or leaf-on conditions. Scientific groups such as the National Ecological Observatory Network (NEON) and NASA's Goddard LiDAR, Hyperspectral and Thermal Imager (G-LiGHT) acquire leaf-on LiDAR, while the United States Geological Survey's National Elevation Dataset (NED) acquires data with leaf-off specifications. For these reasons, models developed using one LiDAR acquisition are often not applicable to another, prohibiting regional LiDAR modeling without unusually consistent LiDAR datasets [9].

### 1.3. Deep Learning

LiDAR EFI have generally been developed using parametric approaches such as regression or non-parametric approaches such as random forest. Here, we use deep learning to overcome the aforementioned obstacles of feature variation by developing a single model for predicting forest attributes that is applicable across many disparate LiDAR and satellite datasets. Deep learning is a form of machine learning and primarily refers to artificial neural networks of a sufficient complexity so as to interpret raw data without a need for human-derived explanatory variables. These differ from simpler neural networks (such as perceptrons), which make estimates using a set of features derived directly from the data (e.g., height percentiles). Recently, deep learning has proven successful at classifying imagery despite varying contextual information, such as light levels and background subject matter [30–32]. We posit that deep learning will improve EFI modeling by identifying useful spatial features in the LiDAR point cloud without the need for human-derived explanatory variables such as height metrics. These features can be complex shapes and gradients in 3-D space that may be less subject to change relative to one another with different acquisition parameters, such as the edges of tree crowns [1].

Here, we implement a spatial deep learning model called a convolutional neural network (CNN). A CNN works by passing a series of moving kernels over spatial data. As the model trains, the weights of those kernels are tuned to identify features that are useful for predicting the dependent variables (such as the edges of objects). Deep CNNs stack many of these moving windows on top of one another, allowing the algorithm to quantify complex features. Our 3-D CNN uses a volumetric window to quantify a LiDAR point cloud that has been binned into voxel-space. The 3-D CNN is thus able to quantify vertical as well as horizontal features and shapes such as tree crowns, providing a level of complexity not captured by height metrics alone.

Early CNNs were developed in the late 1990s and were used to classify hand-written digits [32]. The technique was largely underrepresented in data science until advances in computing power, techniques, and open-source tools popularized them, beginning in 2012 [30]. Since then, CNNs of increasing complexity have consistently outperformed models based on feature extraction for computer vision tasks [33–35]. More recently, CNNs are increasingly being applied to remote sensing problems. For example, 2-D CNNs have proven successful for classifying aerial imagery, hyperspectral, and LiDAR data [36–38]. Although they offer considerable performance improvements in many classification and regression tasks, CNNs do come with some drawbacks: namely, they require far more data to train, access to GPU resources, and often days of training time. However, on inference, they can be applied at speeds that are on-par with other model types [1].

In relation to forestry, some have used segmentation algorithms to isolate individual trees from LiDAR and then used 2-D CNNs to classify tree species [39,40]. Work has also been done using 2-D CNNs to identify individual tree crowns from high-resolution imagery [41,42]. Progress has also been made in adapting CNNs to scan LiDAR point clouds in 3-D space. Similar CNNs that make use of voxels to quantify point clouds have been used to identify household objects and geospatial classification [43,44]. Recently, Qi et al. [45] introduced PointNet, which was designed to interpret LiDAR data without

voxels, although this technique does not make full use of the spatial relationship between neighboring points. In a remote sensing context, Maturana and Scherer [44] used a 3-D CNN to identify helicopter landing zones from LiDAR data. Apart from deep learning, some studies have accounted for disparate LiDAR data by developing different features than height metrics for quantifying forests and by accounting for point cloud variability through principal components analysis of voxels [19,46]. Most similar to this study, Ayrey and Hayes [1] tested a variety of CNN architectures to interpret LiDAR data for the estimation of forest attributes.

### 1.4. Objectives

The first objective of this study is to assess the value of deep learning in developing an EFI, and compare it to traditional approaches for LiDAR modeling. A second objective is to develop a regional EFI over the Acadian/New England Forest region, with a total of 85% coverage of the New England states using publicly-available LiDAR. We assess deep learning's ability to overcome challenges resulting from disparate LiDAR datasets, and we incorporate other remote sensing products such as spectral data, phenology, and disturbance history to improve model accuracy. A final objective is to compare our deep learning-derived mapped estimates of various other stand attributes including biomass, percentage of conifer, and tree stem density to estimates derived via the design-based US national forest inventory program managed by the US Forest Service Forest Inventory and Analysis (FIA). The end result is a series of near wall-to-wall mapped forest inventory estimates of the region, with an accurate assessment of error across space and forest type. This provides forest managers, ecologists, and other scientists in the region with an unprecedented amount of detailed information about the forest.

## 2. Materials and Methods

### 2.1. Forest Attributes

Our goal was to estimate several common forest attributes that may be useful to ecologists, forest managers, and modelers. The complete list is found in Table 1. For brevity, at points throughout this manuscript, we highlight only the results of the aboveground biomass (AGB), percent conifer (PC), and tree count (TC) estimations. All other attributes can also be considered measurements of tree size, density, or species and are often represented by these three attributes.

**Table 1.** A complete list of forest attributes estimated in this study. Note that all estimates were made exclusively on trees greater than 10 cm in diameter (DBH). Estimates were made as quantities per cell.

| Forest Attribute | Units | Description |
|---|---|---|
| Aboveground Biomass (AGB) | kg | Aboveground biomass as calculated by the USFS's FIA component ratio method. |
| Total Biomass | kg | Total woody biomass as calculated by the USFS's FIA component ratio method. |
| Basal Area | $m^2$ | Basal area at breast height. |
| Mean Tree Height | m | Mean height of the trees' apices. Not a measure of mean overall canopy height. |
| Quadratic Mean Diameter | cm | Quadratic mean of diameter at breast height. |
| Volume, Total | $m^3$ | Total inner bark volume of each tree's bole. |
| Volume, Merchantable | $m^3$ | Total merchantable inner bark volume of each tree's bole; starting at 10 cm above ground and ending at a height of 10 cm in diameter. |
| Tree Count (TC) | # | Total number of trees. |
| Percent conifer stems (PC) | % | Percentage of conifer stems. |
| Percent spruce-fir (*Abies-Picea*) | % | Percentage of spruce or fir species stems. |
| Percent *Pinus strobus* | % | Percentage of spruce or fir species stems. |
| Volume of deciduous | $m^3$ | Total inner bark volume of deciduous tree boles. |
| Volume of spruce/fir (*Abies-Picea*) | $m^3$ | Total inner bark volume of spruce or fir tree boles. |
| Volume of *Pinus strobus* | $m^3$ | Total inner bark volume of white pine (*Pinus strobus*) tree boles. |

### 2.2. Training Data

For most applications, deep learning requires very large datasets, with classic open source datasets such as ImageNet numbering in the millions [47]. To meet this requirement,

we combined 13 distinct forest inventories collected at 32 sites (Appendix A Table A1). Within each inventory, all trees greater than 10 cm in diameter (DBH) were stem-mapped with species and DBH recorded. In several inventories, tree heights were measured on only a subset of trees, and so site and species-specific, non-linear height-diameter models were generated using a Chapman–Richards model form to impute tree height using site-specific data [48]. Some inventories were also measured up to 10 years prior to LiDAR acquisition. In instances in which the temporal discrepancy between LiDAR and field data exceeded two years, tree measurements were projected forward in time accordingly using the Forest Vegetation Simulator's Acadian Variant [49].

Total and merchantable volume was estimated using species-specific regional taper equations [50], with a 10 cm upper stem diameter threshold for the latter. Biomass was estimated using the component ratio method developed for the US Forest Service, the Forest Inventory and Analysis (FIA) program [51]. Each of the stem-mapped inventories was aligned visually with the LiDAR to correct for GPS error in plot location and segmented into 10 × 10 m grid cell plots. We selected this cell size to maximize the number of unique plots available, while retaining plots large enough to contain several entire tree crowns. Although highly spatially explicit data are thus provided, this plot size is relatively small compared to those used in other studies and thus may be prone to edge effects.

We accounted for edge effects by using regional diameter-to-crown width equations to project each tree's crown in space [52]. Tree level basal area, biomass, and volume allometry were then multiplied by the proportion of which each tree's crown overlapped the plot. Trees were therefore treated as areas containing biomass, rather than points that could lie on one or another side of a plot boundary [1]. This mimics the method by which the remote sensing instrument measures the trees, as LiDAR imagery has no means of measuring the precise location of a tree's stem. Figure 1 demonstrates this correction. Preliminary testing using a subset of the data indicated that this correction greatly improved model performance, increasing the explained variance by up to 25%.

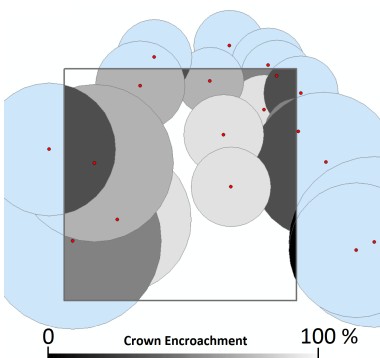

**Figure 1.** Percentage of crown overlap of each tree in and around the 10 × 10 m plot, used as a modifier for that tree's basal area, biomass, and volume. This allowed for the development of models that more closely reflect what is visible to the remote sensing systems, while remaining unbiased across multiple cells.

We also augmented the sample size of our training data by allowing plots to overlap one another by a maximum of 25% and by including multiple LiDAR acquisitions of the same plot, given that the configuration of LiDAR returns always varies between acquisitions. Similar augmentation techniques, such as transforming or rotating input images multiple times and using adjacent still frames of videos, have been successfully used in deep learning for many years [53]. Lastly, we included 500 plots with no trees to allow for better predictions in low-vegetation environments; these were sampled randomly across Northern New England using the 2011 National Land Cover Database [54]. The associated LiDAR point clouds from these plots were then manually checked for trees and discarded if trees were found.

Ultimately, we assembled 24,606 plots for model training and preliminary evaluation. Of these, we randomly withheld 4000 plots, including 1000 for model validation (to determine the optimal stopping point during deep learning training) and 3000 for model testing (used for model comparison and selection). Augmented plots related to withheld plots were removed from the training dataset; thus, the final training set was comprised of 17,432 plots.

*2.3. Remote Sensing Data*

2.3.1. LiDAR

We aggregated 49 public LiDAR datasets from across the region, combining acquisitions of varying pulse density and seasonality. Although much of the pubic LiDAR in the US Northeast is flown leaf-off, we chose to develop models capable of functioning in either state (leaf-off or leaf-on) to allow for potential future integration with leaf-on Canadian Maritime data. The training data ultimately consisted of a 53% to 47% split between leaf-off and leaf-on, respectively. Both leaf-on and leaf-off data were used together to develop each model, and the seasonality of the dataset was included as a binary predictor in the random forest models.

The majority of the LiDAR used for this study was funded and hosted by the US Geological Survey's national 3D Elevation Program (3DEP). These data were captured in leaf-off conditions between 2006 and 2018 at resolutions ranging from 0.5 to 10 pls m$^{-2}$. We also incorporated LiDAR data acquired by NASA Goddard's LiDAR Hyperspectral and Thermal Imager (G-LiHT) as well as the National Ecological Observatory Network (NEON). These data were acquired in leaf-on conditions over several of our training sites with pulse densities ranging from 5 to 16 pls m$^{-2}$. Finally, we incorporated several private LiDAR datasets for training, including one each over the Penobscot Experimental Forest in Maine, Baxter State Park in Maine, and Noonan Research Forest in New Brunswick. Each of these had an average pulse density of 5 to 6 pls m$^{-2}$, where the first two were leaf-off and the third leaf-on. Both pulse density and seasonality were included as model predictors.

2.3.2. Satellite Variables

We also chose to include satellite-derived spectral indices, disturbance metrics, and phenology data in our models for predicting forest attributes. Each of these were spatially contiguous across our study area and have proven useful for predicting forest attributes [55–57]. All satellite data processing was conducted in Google Earth Engine [58].

Using Sentinel-2b data, we generated maps of six spectral vegetation indices: Normalized Burn Ratio (NBR, [59]), Normalized Difference Vegetation Index (NDVI, [60]), Normalized Difference Moisture Index (NDMI, [61]), Red-Edge Chlorophyll Index (RECI, [62]), Greenness Index (GI, [63]) and Triangular Chlorophyll Index (TCI, [63]). Landsat-8 imagery was used to generate three tassel-cap indices (brightness, greenness, and wetness [64]). This imagery was acquired between 2015 and 2017, and imagery from between the 150th and 244th Julian days was used. All images were cloud-masked, and a single median composite was then developed for the study area. Resolutions greater than 10 m were resampled to match our plot size.

We incorporated disturbance history in our models by using Landsat 5–8 and the LandTrendr disturbance detection algorithm [65]. LandTrendr fits a maximum of seven linear segments to the yearly medians of a spectral band within each Landsat pixel. Vertices identify dramatic changes in the spectral characteristics of that pixel in time and often correspond to disturbances. We ran LandTrendr over the greenness and wetness tassel cap indices, as well as NBR. Instances in which LandTrendr identified a vertex in at least two of the three bands within two years of one another were retained as disturbances. We then condensed these data into the year of last disturbance and the magnitude of that disturbance (as a percentage of the vegetation index change).

Finally, we estimated growing season length across our study area using Moderate Resolution Imaging Spectroradiometer (MODIS) data with a resolution of 500 m. This was

derived by subtracting the mean Julian dates of greenness onset from dates of senescence, which has been demonstrated to correlate well with site quality, thereby aiding models that primarily use tree height to infer tree size. Greenness onset and senescence dates were obtained from the MODIS Landcover Dynamics Program [66].

### 2.4. Deep Learning Modeling

#### 2.4.1. Data Preparation

We first prepared the LiDAR data to be scanned by the 3D-CNN by converting it from a point cloud, with each data point representing an X, Y, and Z value, to volumetric pixels (voxels). A height-normalized point cloud was voxelized by segmenting each $10 \times 10 \times 35$ m space (representing a grid cell) into $40 \times 40 \times 105$ bins and then tallying the number of LiDAR points within each bin. Thus, each voxel represented a space of $25 \times 25 \times 35$ cm on our plot. We used vertically rectangular voxels to reduce dimensionality and retain horizontal features. Voxel size was determined through the qualitative testing of several size configurations using a reduced model form. Ultimately, the voxel data took the form of a 3D tensor, over which the kernels of a CNN could be passed. Although CNNs often perform better and train faster when applied to standardized data [30], we attempted several standardization techniques (e.g., z-score, prescience/absence) and found no such improvement.

#### 2.4.2. Deep Learning Model Architecture

Our deep learning model architecture was based loosely on Google's Inception-V3 (Szegedy et al., 2016), which was determined by Ayrey and Hayes (2018) to be better suited for forest estimation than several other commonly-used CNN architectures [1,67]. The underlying model-form was converted to interpret 3D data, and care was taken to maintain a similar proportional dimensionality to the original model (designed to interpret images with a resolution of $224 \times 224$ pixels). The full model architecture is presented in Figure 2.

Inception-V3 consists of a series of preliminary convolution and pooling layers, followed by inception layers, which consist of a number of convolutions of varying sizes that are passed over the incoming data, each designed to detect different features, and are then concatenated. Inception-V3 consists of nine inception layers back to back, with intermittent pooling to reduce dimensionality. The final inception layer is fed into a fully connected layer for a classification or regression prediction. Each convolution was followed by a rectified linear unit (ReLU) threshold function and batch normalization.

The deep learning model was first trained to estimate only AGB using LiDAR. We used transfer learning to initialize the weights of a more complex model using the weights from the simpler one, which was designed to simultaneously predict all 14 of our forest attributes (Table 1). Each forest attribute was standardized using z-scores, thus placing their values on the same scale. A single loss function was then used to optimize the model to predict all forest variables (Equation (1)), whereby the mean of the squared error of the $k$ standardized forest attributes is summarized to a batch level mean of $n$ training samples.

$$loss = \frac{1}{n} \sum_{i=1}^{n} \frac{1}{k} \sum_{j=1}^{k} (y_i - \hat{y})^2 \tag{1}$$

We included the satellite data as side-channel information by first developing a multilayer perceptron to estimate AGB directly from the satellite variables. We used the weights from this model to initialize a subcomponent of the larger model, which produced a $40 \times 40$ tensor that was then concatenated onto the LiDAR voxels [68].

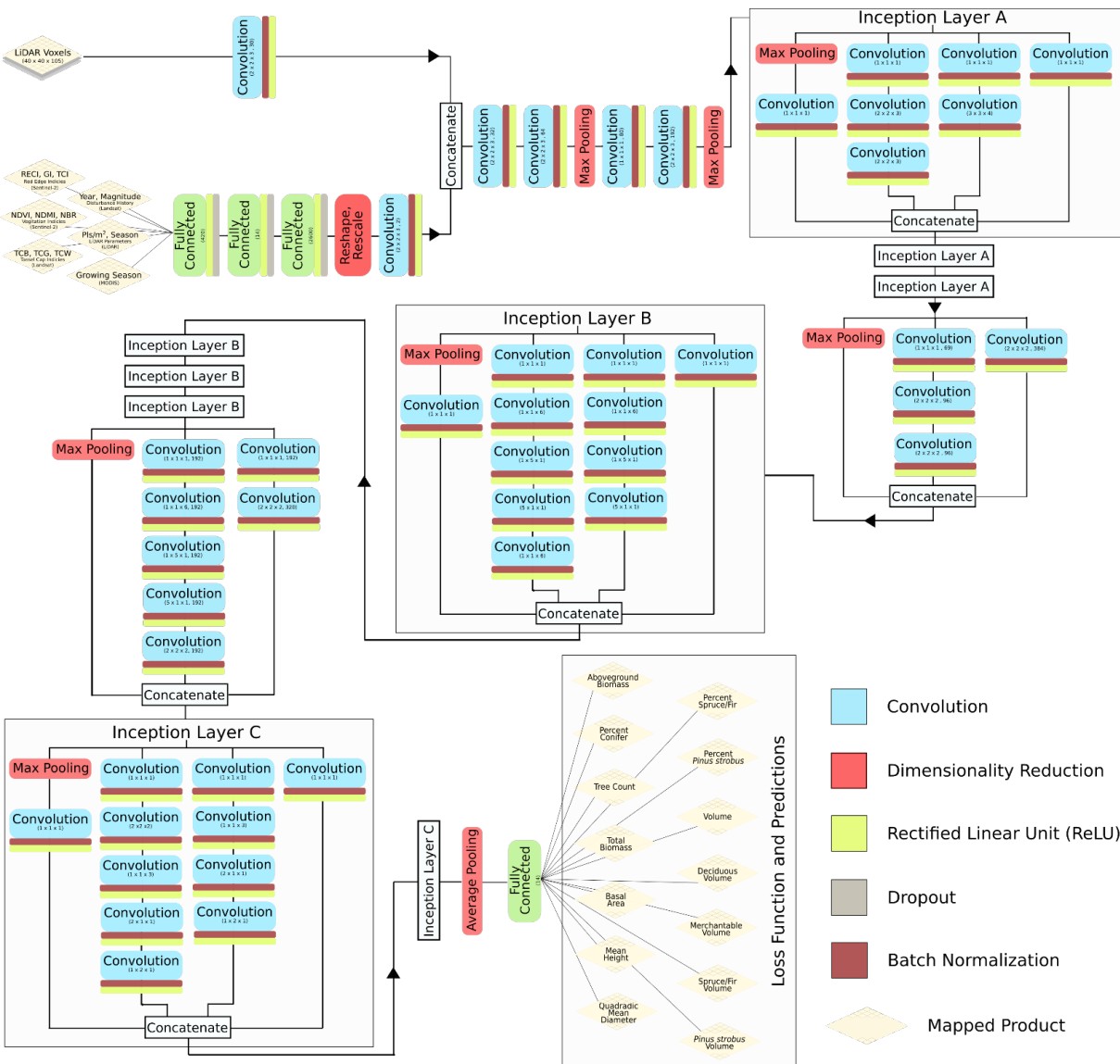

**Figure 2.** The full architecture of the Inception-VS convolutional neural network used to predict forest attributes from LiDAR and satellite data.

### 2.4.3. Deep Learning Model Training

Deep learning models were developed in Python using Google's Tensorflow version 1.15 [69]. The model training process took place in three stages using transfer learning to build upon each stage (1) a AGB model using only LiDAR, (2) a model predicting all 14 attributes using only LiDAR, and (3) a model predicting all 14 forest attributes using LiDAR and satellite metrics. The training time for the first stage was approximately five days using an NVIDIA Tesla k80; the following stages were trained more rapidly. This lengthy multi-step training process made cross-fold validation highly impractical.

### 2.5. Traditional Modeling

We developed traditional models using the standard suite of LiDAR height metrics, derived using the Rlidar package [18]. This package produces a series of summary statistics of LiDAR return heights and proportions above certain height thresholds. We discarded metrics that made use of LiDAR intensity and return counts, as these could not be normalized between the different LiDAR acquisitions. We filtered out points lower than 0.5 m above ground and used a 2 m threshold for many of the proportional metrics. Previous studies in the region have used similar cutoffs [70]. We also included the aforementioned

satellite-derived metrics, as well as pulse density and seasonality. In total, 41 covariates were derived from the LiDAR and satellite data.

Random forest imputation in regression mode was used to model each of the forest attributes [71]. Other studies conducted on subsets of our dataset have demonstrated that this modeling technique outperforms linear mixed-effects modeling [1,11]. We used a Variable Selection Using Random Forest to eliminate unimportant predictors [72]. Each of the models corresponding to the 14 forest attributes had different metrics. The number of metrics used ranged between 5 (deciduous volume) and 27 (tree count). We noticed minor improvements in model performance for each of the 14 models following variable selection. New models were then developed using 2000 decision trees and one-third variable selection at each node-split. These hyper-parameters were fine-tuned using a subset of the data. The random forest models were trained and validated using the withheld test plots. Although accuracy can be assessed using out-of-bag sampling, we used the same validation scheme as the deep learning models due to data augmentation and consistency.

### 2.6. Validation

The training, validation, and testing data derived from the 13 individual forest inventories are likely not fully representative of the landscape, leading to problems with spatial autocorrelation at the regional scale. We therefore performed two phases of validation. The first phase of validation made use of the 4000 withheld plots. This was used for model comparisons between deep learning and traditional modeling and to settle on the final model form.

The second phase of validation made use of an independent dataset and was used to assess the performance of the best model from the first phase once it had been applied across the New England landscape. For this phase, we used the United States Forest Service's FIA national inventory plot data as a ground-truth for verification [73]. We made use of unfuzzed plot locations to validate the maps once they were created, sending them to an authorized party to extract uncertainty estimates. These consisted of approximately 7500 stratified-random plots with a nested sampling design within the states of Connecticut, Maine, Massachusetts, New Hampshire, Rhode Island, and Vermont. We used these to determine map error and bias, assess spatial autocorrelation across the landscape, and compare our inventory estimates to FIA county-level estimates. We assessed errors in Connecticut and Rhode Island separately as their forests increasingly represent a Mid-Atlantic forest type that is not fully represented by the training data. We removed buildings from our maps using a building mask of the United States developed by Microsoft's Bing Maps Team using high resolution imagery [74].

The FIA plots consist of a nested plot design that includes four 7.3 m radius subplots placed 36.6 m away from one another. The subplots have an area of 168 m$^2$, while the entire FIA plot taken as an aggregate has an area of 672 m$^2$. The individual subplot measurements were more affected by errors in plot location, as these were more subject to intra-canopy variability. A preliminary finding that the center plots (on which the GPS point is taken) produced a lower error than the subplots reinforced this conclusion. The aggregate plot-level measurements were less prone to location errors but did not necessarily represent the entire range of variability that one would expect in a 10 m grid cell. Validation plots require roughly the same area as the grid cells being validated so that each have a similar range in values. We therefore assessed accuracy at a subplot and plot level but used the plot-level errors to perform additional analyses. This is roughly equivalent to assessing errors were the map re-sampled to a 20 m resolution.

We decided not to use FIA plots for model training for several reasons: (1) the FIA nested plot design was not compatible with our edge correction technique; (2) FIA plot locations are imprecise and recorded with a consumer-grade GPS and frequently have an error greater than the size of our cells; and (3) FIA plots are relatively small in size (<0.01 ha), which can make linking them to remote sensing difficult.

## 3. Results

### 3.1. Phase One Validation and Model Comparison

The first validation phase made use of withheld plots to compare 14 random forest models using height and satellite metrics to two Inception-V3 CNNs. The first CNN made use of only the LiDAR point cloud, while the second made use of the LiDAR point cloud and satellite metrics. Results in terms of RMSE and bias are displayed in Table 2. With respect to RMSE, both CNNs outperformed random forest in predicting 12 out 14 (85.7%) forest attributes (the two exceptions being predictions of percent conifer and percent spruce-fir). In terms of absolute bias, random forest outperformed both CNNs in half or 7 out 14 or 50% of the metrics. We note that in many comparisons of bias, the absolute difference between models was negligible.

The comparison between CNNs with and without satellite metrics illustrated that the satellite metrics consistently improve the final model's performance. For all forest attributes, the CNN with satellite metrics outperformed those without. The CNN without satellite metrics had less absolute bias in predicting 3 out of 14 (21.4%) attributes. Despite this, the proportional improvement in terms of both RMSE and bias after the addition of satellite metrics was relatively small and so may not justify the additional complexity.

**Table 2.** Results in terms of RMSE, RMSE as a percent of mean (%), and bias of three models. Traditional random forest models trained using LiDAR height and satellite metrics, an Inception-V3 CNN model trained using only the LiDAR point cloud, and an Inception-V3 CNN model trained using the LiDAR point cloud and satellite metrics. The best results achieved are highlighted in green.

| | Traditional Modeling with Satellite Metrics | | CNN without Satellite Metrics | | CNN with Satellite Metrics | |
|---|---|---|---|---|---|---|
| | RMSE (%) | Bias (%) | RMSE (%) | Bias (%) | RMSE (%) | Bias (%) |
| AGB * ($Mg\,ha^{-1}$) | 48.5 (29.4) | −1.2 | 34.5 (20.9) | 1.3 | 33.2 (20.1) | −1.5 |
| PC (%) | 13.3 − | −0.1 | 15.7 − | −2.3 | 14.2 − | −1.7 |
| TC (#) | 2.51 (37.2) | −0.01 | 2.10 (31.1) | −0.6 | 1.75 (26.1) | −0.03 |
| BIOTOT ($Mg\,ha^{-1}$) | 58.2 (29.1) | 1.4 | 41.5 (20.8) | 1.2 | 40.0 (18.6) | −2.58 |
| BA ($m^{-2}$) | 0.083 (24.5) | −0.001 | 0.065 (19.3) | −0.004 | 0.063 (20.0) | −0.009 |
| HT (m) | 2.6 (15.2) | 0.2 | 2.1 (12.0) | 0.6 | 1.5 (9.1) | −0.07 |
| QMD (cm) | 5.9 (23.9) | 0.14 | 4.3 (17.2) | 0.2 | 3.8 (15.3) | 0.1 |
| PSF % | 8.3 − | 0.1 | 10.8 − | −3.6 | 9.5 − | −0.2 |
| PWP % | 8.6 − | −0.1 | 6.7 − | 0.1 | 6.6 − | −0.3 |
| VOL ($m^{-3}$) | 0.81 (27.5) | −0.019 | 0.580 (19.7) | −0.013 | 0.558 (18.9) | −0.022 |
| VOLM ($m^{-3}$) | 0.751 (27.8) | −0.016 | 0.460 (20.1) | −0.012 | 0.524 (19.3) | −0.033 |
| VOLD ($m^{-3}$) | 0.756 (60.6) | 0.002 | 0.460 (36.9) | −0.023 | 0.446 (35.8) | −0.017 |
| VOLSF ($m^{-3}$) | 0.31 (91.4) | 0.007 | 0.227 (67.1) | 0.016 | 0.238 (70.2) | −0.017 |
| VOLWP ($m^{-3}$) | 0.483 (94.2) | −0.016 | 0.388 (67.1) | −0.075 | 0.383 (74.6) | −0.039 |

\* Forest attribute abbreviations are as follows: AGB = aboveground biomass, PC = percent conifer, TC = tree count, BIOTOT = total biomass, BA = basal area, HT = mean tree height, QMD = quadratic mean diameter, PSF = percent *Abies-Picea*, PWP = percent *Pinus strobus*, VOL = inner bark volume, VOLM = merchantable volume, VOLD = deciduous volume, VOLSF = *Abies-Picea* volume, VOLWP = *Pinus strobus* volume.

### 3.2. Phase Two Validation (CNN Model Only)

We performed the second phase of validation after using the best performing model (CNN with satellite metrics) to map each of the forest attributes across New England. In the second phase, each of the mapped attributes was validated using independent FIA plots. Table 3 displays the results of this validation in terms of RMSE, RMSE as a percent of mean (nRMSE), and bias at both the subplot and plot level. We assessed error in northern and southern New England separately because our training data were located only in northern New England. We did not assess the performance of random forest in this phase because these models were not applied at a regional level; such an effort would have been computationally costly and was not considered necessary given our model selection process in the first phase. Model error at the subplot level was considerably higher than error at the plot level for each forest attribute. This is to be expected given that smaller areas are more likely to contain extreme values and the subplot values are more likely to be affected by the small plot size and GPS inaccuracy.

Results of the second phase of validation indicated that two phases were in fact necessary to obtain a more representative assessment of regional model performance. Plot-level RMSE was poorer than the RMSE obtained from the first phase of validation in all forest attributes, indicating that the withheld plots likely did not represent regional landscape heterogeneity. For some forest attributes, this difference was relatively minor. The performances of the stem density, mean height, and species estimates were notably worse in the second phase of validation. The error of each of these values increased between 35–120% from that observed in the first phase of validation.

Overall, the error and bias of attributes representing tree size were lower than those representing species or stem density. Aboveground biomass, total biomass, basal area, mean tree height, QMD, inner bark, and merchantable volume all had a plot-level nRMSE of less than 50% in northern New England. In contrast, tree count had an nRMSE of 57%. Model performance was also poorest in volume estimates of species groups. Estimates of spruce-fir and *Pinus strobus* volume both had nRMSEs above 150% and could generally be considered not useful. We did not assess the nRMSE of the species attributes that were quantified as percentages. The RMSE and bias of percent spruce-fir and percent *Pinus strobus* were lower than that of percent conifer, likely because their average values were smaller. Qualitatively, the maps of percent conifer appeared better, with the other species estimates suffering from artifacts owing to LiDAR flight lines and local biases.

We also assessed the model performance of AGB, PC, and TC in northern New England spatially and by plotting their predicted versus observed values. Figure 3 illustrates the plot level bias of each of these forest attributes. We note that the AGB bias appears to be fairly evenly distributed across the landscape, with consistent model biases not immediately apparent. Negative PC bias appears to be clustered mostly in eastern Maine, where a greater number of conifers are likely to be found, indicating that the model is underestimating in areas of proportionally higher conifers. Likewise, negative bias in Vermont is an indication that the model is underestimating in areas with proportionally fewer conifers. Tree count followed a similar trend where greater negative bias was encountered in more northern areas, which corresponds to the greater stem density found in more intensively-managed industrial forests.

**Table 3.** Results in terms of RMSE, RMSE as a percent of mean (%), and bias of the Inception-V3 model using FIA plot data for validation. Assessments were made using the FIA subplots (roughly corresponding to 10 m cell validation) and the FIA plots taken as an aggregate (roughly corresponding to 20 m accuracy).

| | | | AGB * (Mg ha$^{-1}$) | PC (%) | TC (#) | BIOTOT (Mg ha$^{-1}$) | BA (m$^{-2}$) | HT (m) | QMD (cm) | PSF (%) | PWP (%) | VOL (m$^{-3}$) | VOLM (m$^{-3}$) | VOLD (m$^{-3}$) | VOLSF (m$^{-3}$) | VOLWP (m$^{-3}$) |
|---|---|---|---|---|---|---|---|---|---|---|---|---|---|---|---|---|
| Northern New England (MA, ME, NH, and VT) | FIA Plot- Level Assessment | RMSE | 36.9 | 19.2 | 3.0 | 44.5 | 0.08 | 3.3 | 5.2 | 17.0 | 13.2 | 0.625 | 0.578 | 0.416 | 0.298 | 0.454 |
| | | (%) | 46 | – | 57 | 46 | 45 | 30 | 31 | – | – | 46 | 48 | 58 | 123 | 235 |
| | | Bias | 1.18 | 1.89 | −0.541 | 2.26 | 0.03 | 1.35 | 0.38 | −1.96 | −4.25 | 0.069 | 0.094 | 0.006 | −0.035 | 0.034 |
| | FIA Subplot- Level Assessment | RMSE | 57.0 | 25.6 | 4.67 | 68.6 | 0.12 | 4.4 | 7.7 | 21.8 | 15.0 | 0.994 | 0.922 | 0.622 | 0.389 | 0.587 |
| | | (%) | 71 | – | 89 | 71 | 67 | 41 | 46 | – | – | 76 | 79 | 87 | 161 | 365 |
| Southern New England (CT and RI) | FIA Plot- Level Assessment | RMSE | 44.3 | 19.2 | 3.0 | 53.1 | 0.07 | 2.8 | 10.0 | 7.8 | 0.557 | 0.602 | 0.416 | 0.274 | 0.796 | |
| | | (%) | 39 | – | 58 | 39 | 37 | 37 | 23 | – | – | 49 | 52 | 58 | 5326 | 514 |
| | | Bias | −3.46 | 1.89 | −0.54 | −3.42 | 0.013 | 1.33 | 4.4 | 0.48 | 0.235 | 0.09 | 0.006 | 0.114 | 0.231 | |

* Forest attribute abbreviations are as follows: AGB = aboveground biomass, PC = percent conifer, TC = tree count, BIOTOT = total biomass, BA = basal area, HT = mean tree height, QMD = quadratic mean diameter, PSF = percent *Abies-Picea*, PWP = percent *Pinus strobus*, VOL = inner bark volume, VOLM = merchantable volume, VOLD = deciduous volume, VOLSF = *Abies-Picea* volume, VOLWP = *Pinus strobus* volume.

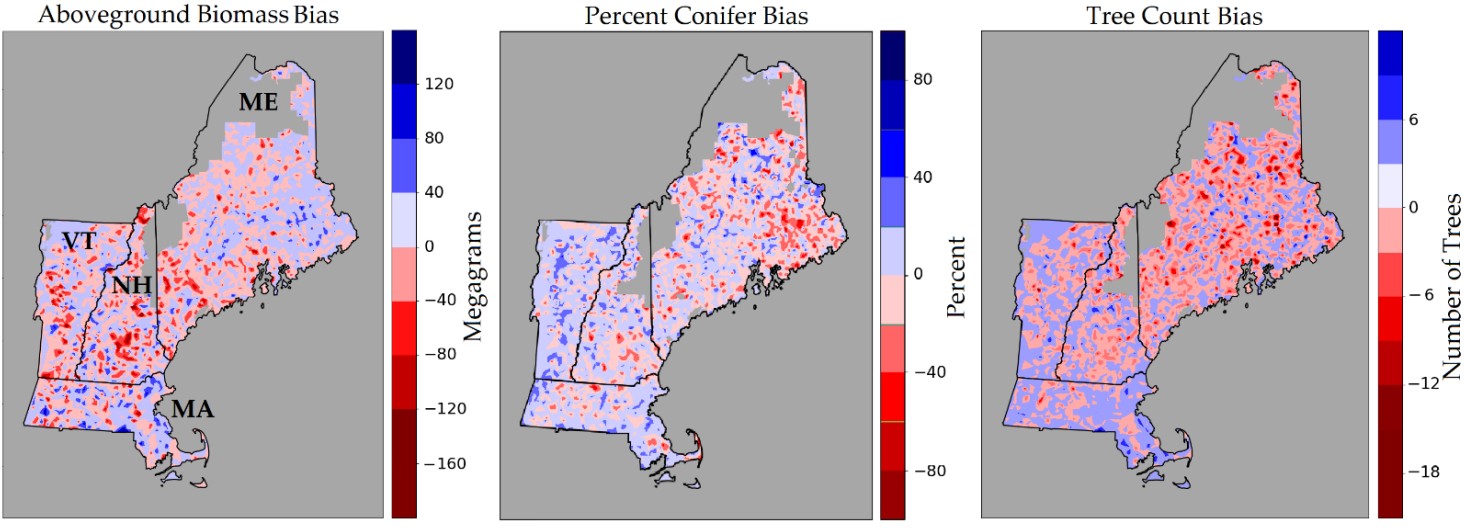

**Figure 3.** FIA plot-level bias is plotted for three of the forest attributes mapped using the CNN. Red areas denote model underestimation, blue areas denote model overestimation. In blank areas, LiDAR has not yet been acquired. Fuzzed plot locations were used to develop these maps.

These trends can also be observed in the predicted-versus-observed plots (Figure 4). Biomass residuals fall relatively tightly along the 1:1 line, with little to no attenuation observed at higher biomass values. Percent conifer residuals seemed to indicate a tendency to overestimate in low conifer environments and underestimate in high conifer environments. Finally, tree count residuals appeared to follow the 1:1 line in low-medium density conditions but often severely underestimated tree count in high-density conditions.

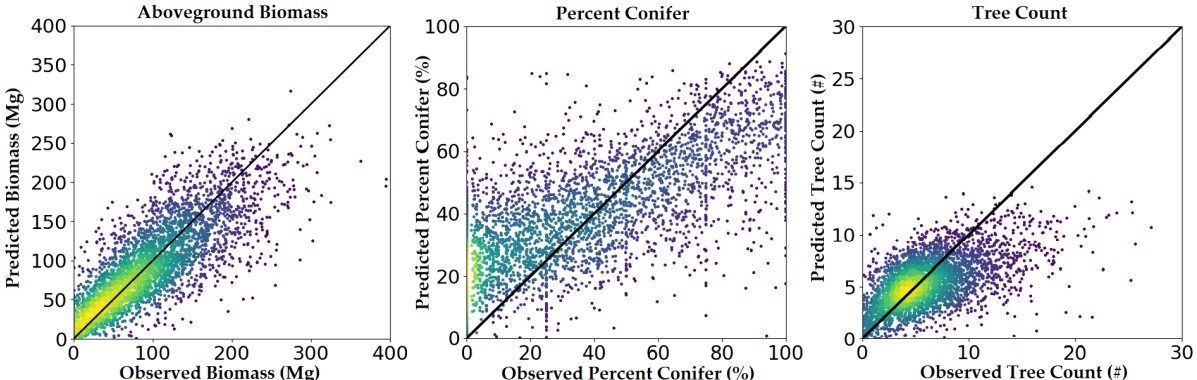

**Figure 4.** Predicted versus observed plots using FIA plot-level validation. Warmer colors represent greater numbers of observations.

We assessed model performance across different LiDAR datasets by plotting the northern New England plot-level bias as a function of pulse density (Figure 5). We used loess regression to fit a moving trendline to the data using 75% of observations to smooth the line at each value. No biases stemming from pulse density were apparent in this visualization, with the trend lines for biomass and tree count consistently near zero, and the trend line for percent conifer showing a slight positive bias, but with no apparent trend with pulse density. Nevertheless, banding was visible in the percent conifer and tree count maps in regions of very low pulse density (not shown). Bands appeared to follow trends in average scan angle along each flight line and may have been a function of pulse density and scan angle combined. Unfortunately, we did not map the mean scan angle across the landscape a-priori as we did for pulse density.

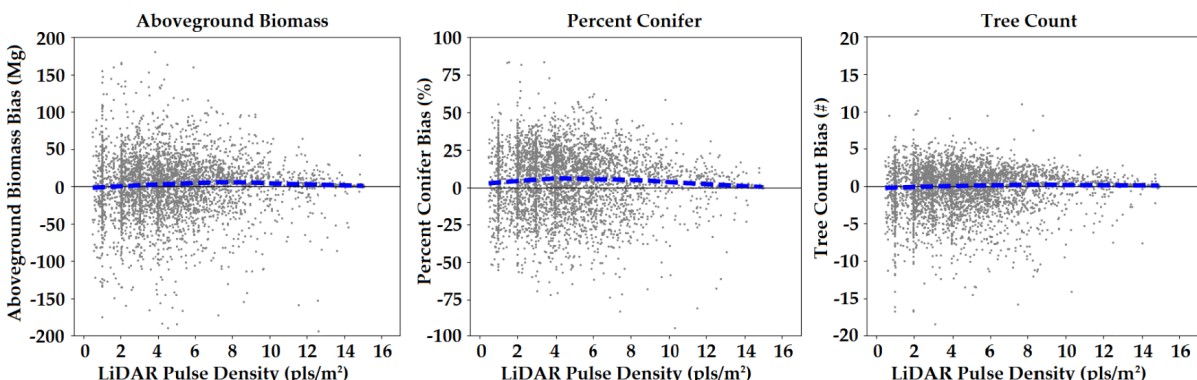

**Figure 5.** Bias by pulse density in northern New England using FIA plot-level validation. The dashed blue line is a loess regression fit of the data.

### 3.3. County-Level Comparisons

With the region mapped using the best performing CNN, we compared county-level estimates derived by summing the values of our map with FIA design-based estimates (Table 4). We chose the 38 counties in northern New England with complete LiDAR coverage. Initially, we used all FIA plots within a county measured within two years

of the LiDAR acquisition. We discovered, however, that a large number of FIA plots without measured trees were located in suburban environments with trees. This resulted in underestimates by the FIA data of each forest attribute, so plots that were denoted as having no trees that fell within semi-forested suburbs were removed after manual aerial photo interpretation.

**Table 4.** County-level estimates of total aboveground biomass (AGB), percent conifer (PC), and tree count (TC) are compared using the FIA's design-based sampling and summations of our forest inventory maps. Blue denotes mapped estimates that fell within the 95% confidence interval of the FIA's estimate, yellow denotes estimates that fell within 97.5%, and red denotes values that estimates that differed from the FIA.

| State | County | FIA AGB (Petagrams) | CNN AGB (Petagrams) | FIA PC (%) | CNN PC (%) | FIA TC (Millions) | CNN TC (Millions) |
|---|---|---|---|---|---|---|---|
| Maine | Cumberland | 21.4 ± 3.5 | 18.8 | 34.4 ± 5.7 | 26.7 | 106.6 ± 17.4 | 89.8 |
| Maine | Hancock | 29.3 ± 3.5 | 29.6 | 55.6 ± 5.1 | 43.3 | 277.6 ± 32.1 | 244.2 |
| Maine | Kennebec | 17.3 ± 3.0 | 16.7 | 30.0 ± 5.4 | 28.9 | 112.4 ± 19.5 | 92.9 |
| Maine | Knox | 7.5 ± 2.1 | 5.5 | 42.7 ± 9.2 | 30.7 | 52.5 ± 13.1 | 35.1 |
| Maine | Lincoln | 9.1 ± 2.1 | 8.3 | 35.0 ± 8.3 | 32.4 | 55.0 ± 14.3 | 48.6 |
| Maine | Penobscot | 56.9 ± 4.6 | 56.2 | 51.1 ± 3.6 | 43.3 | 585.1 ± 42.0 | 452.1 |
| Maine | Piscataquis | 65.0 ± 5.2 | 61.1 | 50.9 ± 3.5 | 46.3 | 655.0 ± 50.0 | 528.0 |
| Maine | Sagadahoc | 6.7 ± 1.5 | 5.18 | 39.5 ± 12.7 | 33.5 | 40.0 ± 10.0 | 29.7 |
| Maine | Waldo | 16.0 ± 2.3 | 11.3 | 38.7 ± 5.4 | 33.9 | 124.1 ± 18.5 | 71.1 |
| Maine | Washington | 42.8 ± 3.9 | 43.1 | 50.9 ± 3.7 | 46.5 | 422.4 ± 32.5 | 404.3 |
| Maine | York | 25.9 ± 3.3 | 22.3 | 30.8 ± 5.3 | 27.5 | 140.0 ± 17.0 | 105.1 |
| Massachusetts | Barnstable | 4.5 ± 1.5 | 4.4 | 36.5 ± 13.0 | 24.0 | 36.3 ± 12.8 | 41.3 |
| Massachusetts | Berkshire | 34.4 ± 4.4 | 32.6 | 18.7 ± 5.1 | 28.7 | 114.5 ± 15.4 | 126.3 |
| Massachusetts | Bristol | 11.9 ± 3.0 | 11.9 | 15.5 ± 5.3 | 23.9 | 53.0 ± 15.6 | 56.4 |
| Massachusetts | Dukes | 0.9 ± 0.5 | 0.85 | 20.5 ± 20.2 | 18.9 | 6.3 ± 3.0 | 8.7 |
| Massachusetts | Essex | 14.0 ± 2.1 | 11.2 | 23.6 ± 5.2 | 18.4 | 57.7 ± 12.1 | 39.3 |
| Massachusetts | Franklin | 24.0 ± 3.8 | 27.3 | 31.4 ± 5.9 | 30.0 | 86.9 ± 14.2 | 93.1 |
| Massachusetts | Middlesex | 20.8 ± 4.3 | 17.7 | 23.5 ± 8.1 | 21.6 | 61.0 ± 15.3 | 65.82 |
| Massachusetts | Nantucket | 0.3 ± 1.2 | 0.1 | 0.0 | 7.0 | 2.5 ± 8.8 | 0.8 |
| Massachusetts | Norfolk | 9.7 ± 3.1 | 7.9 | 14.9 ± 8.8 | 21.8 | 36.7 ± 11.7 | 33.7 |
| Massachusetts | Plymouth | 13.7 ± 3.7 | 14.4 | 34.0 ± 8.6 | 27.2 | 58.8 ± 14.0 | 67.3 |
| Massachusetts | Suffolk | NO PLOTS | 0.3 | NO PLOTS | 10.7 | NO PLOTS | 2.1 |
| Massachusetts | Worcester | 44.6 ± 5.8 | 41.2 | 22.9 ± 4.8 | 24.0 | 145.0 ± 18.9 | 154.4 |
| New Hampshire | Belknap | 10.5 ± 2.4 | 11.1 | 29.2 ± 8.6 | 25.5 | 50.8 ± 11.6 | 45.4 |
| New Hampshire | Cheshire | 26.5 ± 3.2 | 25.6 | 29.1 ± 5.6 | 31.8 | 105.1 ± 14.8 | 98.3 |
| New Hampshire | Hillsborough | 25.8 ± 4.0 | 23.5 | 28.8 ± 5.5 | 27.3 | 102.8 ± 15.7 | 92.3 |
| New Hampshire | Merrimack | 32.5 ± 4.4 | 24.6 | 31.8 ± 4.8 | 27.7 | 134.8 ± 17.1 | 99.8 |
| New Hampshire | Rockingham | 19.1 ± 3.9 | 17.9 | 32.2 ± 6.3 | 22.3 | 72.7 ± 14.8 | 72.7 |
| New Hampshire | Strafford | 9.4 ± 2.4 | 9.8 | 25.5 ± 8.1 | 27.5 | 38.4 ± 10.2 | 42.4 |
| New Hampshire | Sullivan | 16.1 ± 2.6 | 17.0 | 30.9 ± 7.2 | 33.6 | 74.5 ± 12.5 | 72.3 |
| Vermont | Caledonia | 15.2 ± 3.1 | 14.8 | 35.6 ± 8.2 | 33.1 | 85.4 ± 16.7 | 76.0 |
| Vermont | Essex | 10.6 ± 1.6 | 14.0 | 23.6 ± 5.2 | 39.5 | 76.1 ± 9.2 | 93.0 |
| Vermont | Lamoille | 13.7 ± 2.1 | 12.4 | 23.2 ± 8.2 | 35.6 | 60.1 ± 9.5 | 62.4 |
| Vermont | Orange | 18.2 ± 3.2 | 19.2 | 32.1 ± 8.2 | 30.8 | 78.3 ± 14.2 | 79.2 |
| Vermont | Rutland | 30.8 ± 3.0 | 25.7 | 19.0 ± 4.3 | 26.7 | 116.5 ± 11.8 | 110.8 |
| Vermont | Washington | 18.1 ± 2.9 | 19.6 | 29.4 ± 6.6 | 33.6 | 85.9 ± 13.2 | 91.0 |
| Vermont | Windham | 28.0 ± 3.4 | 27.6 | 26.1 ± 6.2 | 33.1 | 102.4 ± 12.2 | 103.7 |
| Vermont | Windsor | 30.8 ± 4.3 | 30.9 | 19.2 ± 5.5 | 30.0 | 112.5 ± 16.8 | 117.0 |

Our aboveground biomass predictions fell within the 95% confidence interval of the FIA's estimate in 31 out 38 counties (81.5%) and within a 97.5% confidence interval in 33 out 38 counties (86.8%). Across these counties, FIA estimated 4% more biomass than our map, which is to be expected given that our maps frequently had gaps between LiDAR acquisitions and occasionally had missing LiDAR tiles. The FIA's lack of urban tree sampling also likely played a large role in this discrepancy. Eight of the counties were classified by the US Census Bureau as having an urban population greater than 50%. In these urbanized counties, FIA estimates were an average of 13% lower than ours when including empty plots in suburban forested areas and 11% greater than ours after these plots were removed.

In estimating percent conifer, 25 out 38 (65.7%) of our estimates fell within the 95% confidence interval of the FIA's estimate. In agreement with the map of residuals, the

percentage of conifers was significantly underestimated in 6 out 11 counties (54.5%) in Maine and significantly overestimated in 4 out 8 counties (50.0%) in Vermont. In 16 out 19 counties (84.2%) in Massachusetts and New Hampshire, our estimate of percent conifer was within the 95% confidence interval of the FIA's estimate. Across the entire landscape, the two estimates were within 0.2% of one another.

County-level stem density estimates fell within the FIA's 95% confidence interval 30 out 38 times (78.9%). Once again, counties in Maine with greater numbers of small trees were more likely to be underestimated. In 5 out 12 Maine counties (41.7%), mapped estimates of stem density were significantly lower than the FIA's estimates. Outside of Maine, 24 out 27 counties (88.9%) had mapped estimates within the 95% confidence interval of the FIA's estimate. This is likely owing to an overall greater stem density in Maine due to greater commercial forest management and a general tendency of boreal forests to be denser.

## 4. Discussion

Our results indicate that 3-D convolutional neural networks (CNNs) can be used to effectively estimate forest attributes from disparate LiDAR and satellite data. These models outperformed random forest models, which are the traditional approach for generating forest inventories from LiDAR. They could also be effectively scaled to make regional-scale, high-resolution maps/estimates, which we demonstrate were often statistically equivalent to traditional ground-based forest inventories.

### 4.1. Model Comparison

Our first objective was to compare LiDAR-derived inventory estimates made using CNNs to estimates made using height metrics and random forest modeling. We assessed this in our first phase of validation, in which several models were developed from training data and assessed using withheld plots. Random forest models trained using traditional height metrics and satellite data nearly always had a greater error than the two CNNs (with and without satellite data) that we trained. This finding corroborates that of Ayrey and Hayes [1], in which 3-D CNNs of varying architectures often outperformed generalized linear models and random forest. These results also indicate that deep learning (CNNs) can be a more effective way of modeling forest attributes from LiDAR data than the traditional approach using LiDAR height metrics.

In the estimation of species (percent conifer and percent spruce-fir), random forest outperformed CNNs. These species estimates likely relied more heavily on satellite spectral data than on LiDAR structural data. We speculate that random forest made better use of the satellite covariates than did our CNN. The CNN was initially trained to scan LiDAR voxels, and the satellite covariates were added afterwards in such a way as to concatenate satellite data onto voxel space. This process may have been less than ideal. Zhou and Hauser [68] outlined several methods for including side-channel data into a CNN. When applied to our data, their methods produced mixed results, and we ultimately settled on our concatenation method.

The CNNs also did not outperform the random forest models in terms of bias. Half of the random forest models had a lower absolute bias than did the CNNs, indicating that both model types performed similarly. We did not observe any notable trends in bias by forest attribute. Overall, we believe that the differences in absolute bias between models were often low enough to be attributed to the random variations of the testing data. Likewise, Legaard et al. [75] indicated that random forest and single-objective support vector regression also produced the lowest total prediction error for estimating tree species abundance in Maine yet produced the greatest systematic error, consistent with the strong attenuation bias of these methods.

The comparison in our first phase of validation between the CNNs with and without satellite metrics highlighted that the CNN benefitted from spectral and disturbance information. The error decreased when estimating every forest attribute with satellite imagery,

and bias generally decreased as well. This improvement was modest, suggesting that even without the satellite metrics, a CNN could be trained to outperform traditional random forest models using height and satellite metrics. We chose not to explore which satellite metrics were most useful, as deep learning models of this size run no risk of overfitting with extraneous predictors. However, such an analysis would be possible through a process similar to random forest's derived importance and might be useful in identifying necessary remote sensing datasets.

Some LiDAR acquisitions have been designed to sample portions of forested regions rather than acquire wall-to-wall coverage. In this context of model-based estimation (where LiDAR data may be used to infer population-level estimates without complete coverage), deep learning may prove impractical for computing uncertainty estimates due to the computational demands of bootstrapping [76]. Studies using model-assisted (or design-based) paradigms may still be able to take advantage of this approach to achieve better accuracy than traditional modeling techniques.

We note several advantages to working with a single deep learning model aside from better performance. Our Inception-V3 CNN took a considerable amount of data and time to train; however, once trained, the model could quickly be applied to large regional LiDAR datasets. A single model predicting all 14 forest attributes presented less of a data-management challenge than 14 separate models. We also suspect that our CNN would be less likely to produce conflicting estimates than would 14 separate unconstrained models (e.g., more merchantable volume than total volume). Other studies have also noted similar benefits of training models with multiple response variables [77,78].

Finally, the field of deep learning is now making use of pre-trained model weights to solve novel problems [79,80]. The rapid retraining of our CNNs indicates that this model can easily be fine-tuned with local data, re-tuned with non-local data, or applied to different problems to save modelers the effort of training a large CNN to interpret voxel space with randomized weights. The weights from our CNN could be used to initialize CNNs with other LiDAR-related objectives, such as individual tree crown segmentation or LiDAR classification.

### 4.2. Assessing Performance

With the final CNN model, we mapped all 14 forest attributes across the study area (Figures 6–8). We assessed map performance with our second phase of validation, which made use of independent FIA plots. We assessed accuracy at a subplot, plot, and county level. Our subplot-level error estimates were consistently quite high. The FIA plot locations in this region are subject to considerable error, and the FIA notes that plot location error can be as high as 100 m (although in practice most plots are located within 12 m of their measured location, [81]). Examining pixel-level accuracy using these subplots was problematic given the high degree of intra-canopy variation present in 10 m pixels. We observed that the center subplot (on which the GPS location was taken) resulted in lower map error, which is a further indication that the locational accuracy of the surrounding subplots is suspect.

Our assessment of plot-level error (the aggregate of the subplots) produced more favorable error results. We achieved nRMSE values of between 30% to 48% for those attributes quantifying tree size, which we consider to be a success given the small size of the grid cells used. We consider estimates of tree count, percent conifer, and deciduous volume to have been made with moderate success, with nRMSE values of 56% to 58%. We consider estimates of *Pinus strobus* and spruce-fir species breakdowns to be a failure, with nRMSEs exceeding 100%. Nevertheless, these maps may be of use to practitioners when aggregated to a coarser resolution and binned into categories.

We assessed model performance at the county level and in space using aboveground biomass, percent conifer, and tree count. Our map of biomass bias across northern New England (Figure 6) appeared to be relatively uniform, indicating that the model represented biomass across the landscape well. Notably, the model did not experience any saturation

of high biomass areas, as is often the case with regional remote sensing studies [82]. Our biomass estimates fell within the 97.5% confidence of FIA biomass estimates in all but four counties. Those four counties appeared to have little in common in terms of forest characteristics, proximity, or human population density. In urban and urbanized counties, FIA estimates of biomass could underestimate or overestimate those from the CNN, depending on sampling design. Retaining supposedly empty FIA plots placed in suburban areas where trees were present in aerial imagery resulted in the FIA underestimating biomass relative to the CNN. This suggests that our maps are better able to quantify urban and suburban biomass. By our estimate, this adds up to an additional 13% biomass in urbanized counties. However, we note that none of our training data made use of urban plots, and few of our training plots had trees grown in the open. This improvement may be a case of any estimate being better than none at all.

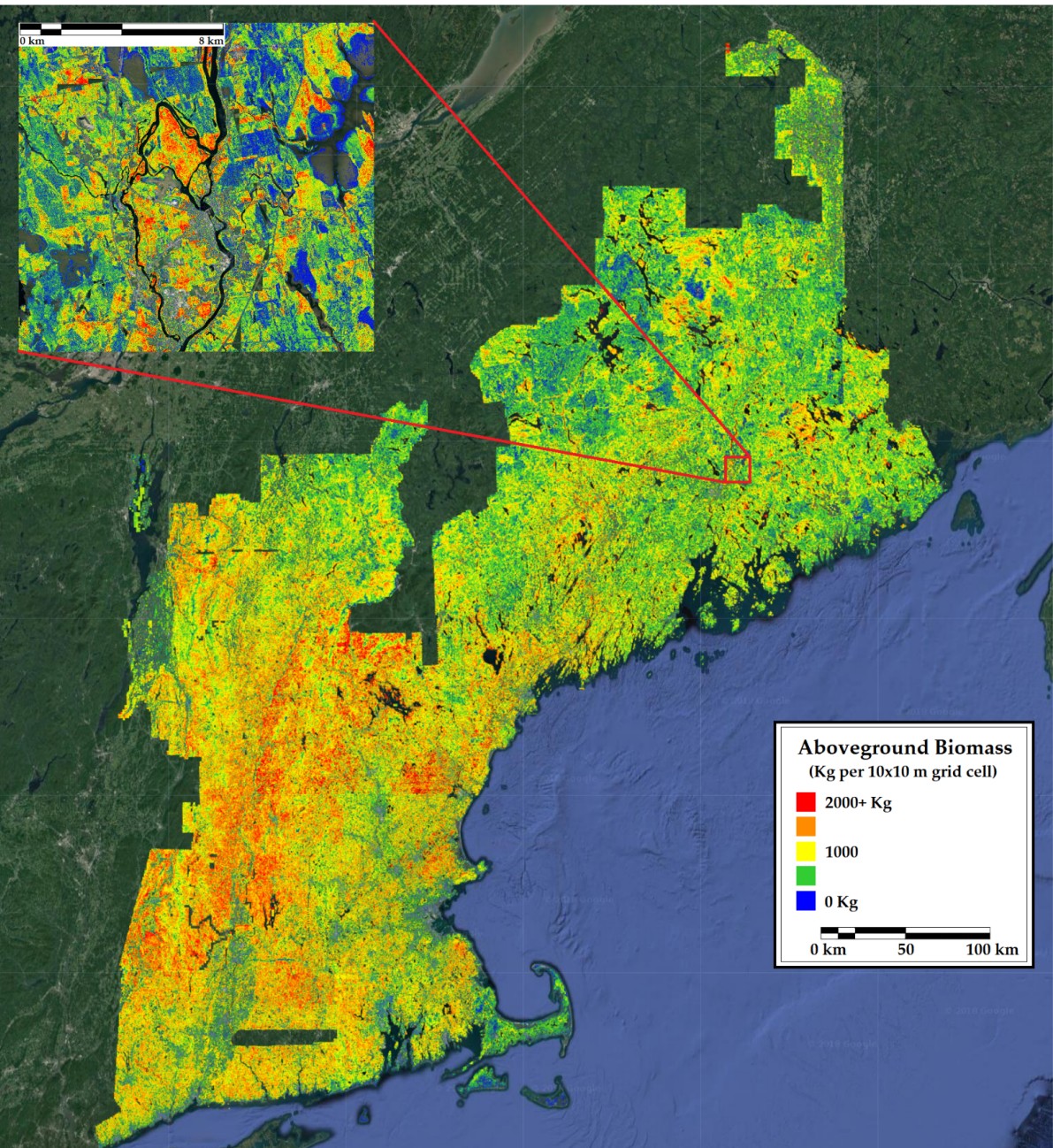

**Figure 6.** A 10 m resolution forest inventory map of aboveground biomass in New England. Included is a 12 km inset of a representative portion of the region.

The map of percent conifer bias (Figure 3) and the county-level comparison (Table 4) showed a systematic underestimation of percent conifer in eastern Maine and an overestimation in Vermont. These areas are inhabited by very high and low proportions of conifers, respectively. The predicted versus observed plot confirms that the CNN underfit the extremes in percent conifer. The map of tree count bias and the count-level comparison was similar to the percent conifer in that there was a consistent underestimation of tree numbers in northern Maine, where stem densities are naturally higher due to the species assemblages and greater harvest intensity resulting in younger forests. The predicted versus observed plot highlights a model saturation in very high density forests. Unlike percent conifer, no consistent underestimation of tree count was observed in lower density areas. Intuitively, one might expect this result given that the structures of very dense forest stands resemble one another despite different stem densities (e.g., a point cloud representing a stand with 2000 trees per ha looks very similar to one representing 2500 trees per ha). Satellite indices are also sometimes prone to the same type of saturation at very high stem densities [83].

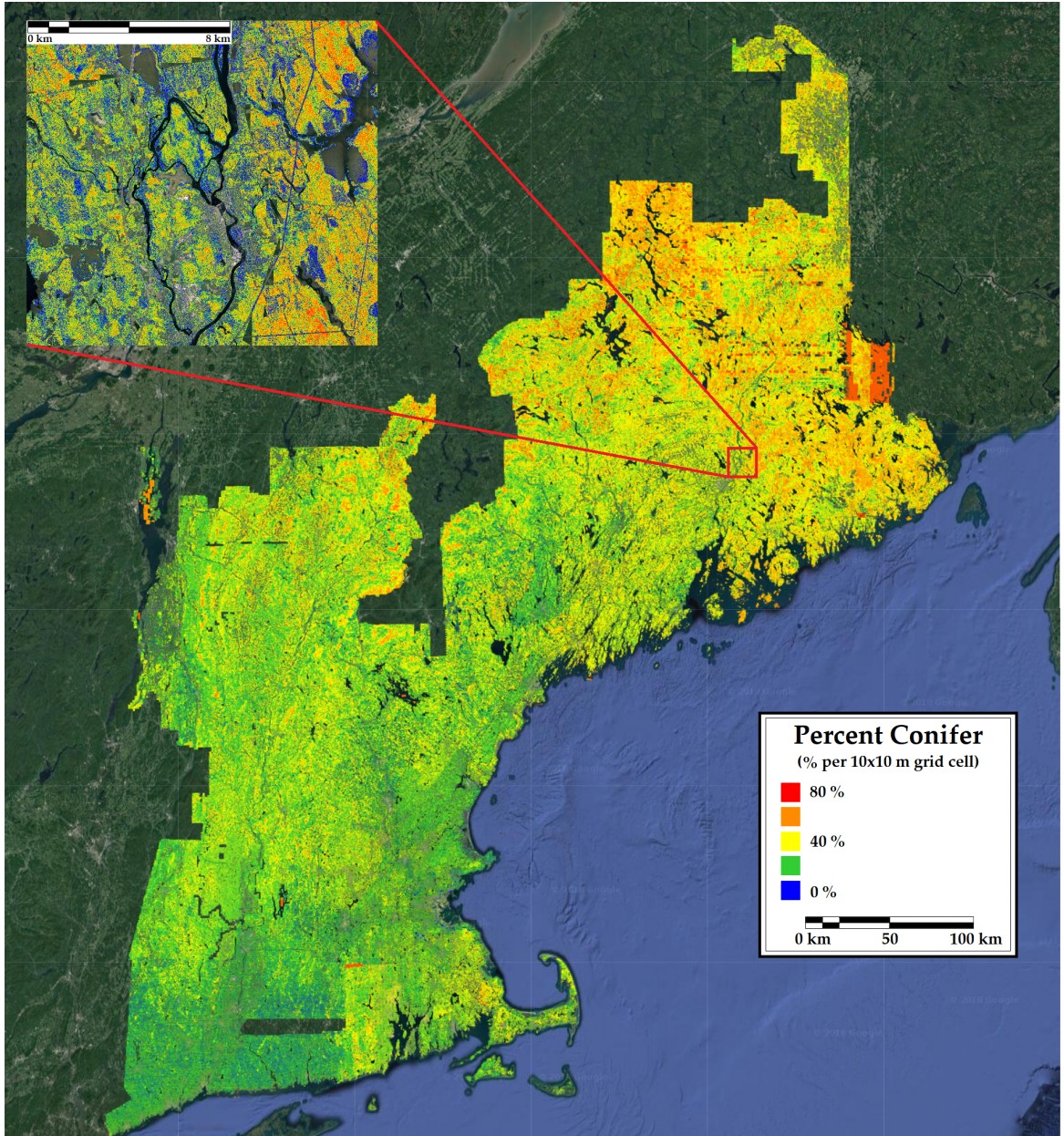

**Figure 7.** A 10 m resolution forest inventory map of percent conifer in New England. Included is a 12 km inset of a representative portion of the region.

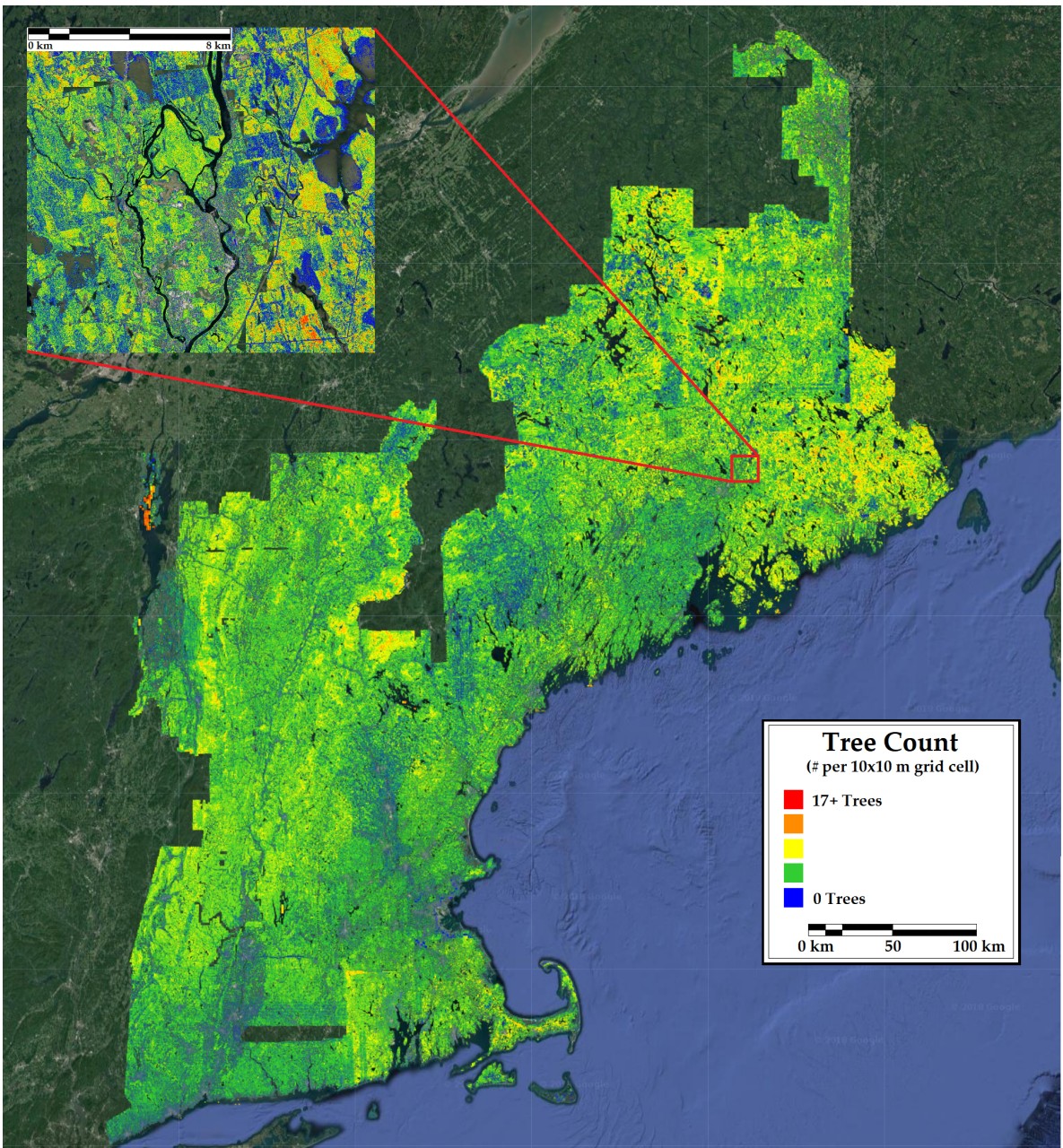

**Figure 8.** A 10 m resolution forest inventory map of tree count in New England. Included is a 12 km inset of a representative portion of the region.

The CNN model can be summarized as being an effective predictor of attributes closely related to tree size (e.g., aboveground biomass), being moderately effective at predicting attributes related to tree density and percent conifer, and being a poor predictor of attributes related to species groupings. Previous studies modeling forest attributes using LiDAR have likewise had more difficulty in estimating stem density and species [70,77,84,85]. We present the following possible explanations for the model's under-performance here:

1.  Although we incorporated satellite spectral indices useful for species estimation, the model architecture may not have made full use of them.
2.  Stem density was often underestimated in high density stands, but qualitatively, the maps seemed to suffer from banding in areas with low pulse density LiDAR ($<3$ pls/m$^2$). The model may have made use of horizontal structural features in

the canopy that could not be resolved in low density LIDAR. Ayrey and Hayes [1] determined that 3-D CNNs make use of the horizontal canopy structure, such as the edges of tree crowns.

3. In re-examining our loss function, we find that half of our forest attributes were in some way related to tree size, while only one attribute estimated stem density. Thus, our unweighted loss function may have inadvertently favored attributes estimating tree size, resulting in a model that identified features in the LiDAR data that were more predictive of size, rather than density or species.

### 4.3. Mapping Errors

The regional maps of forest attributes suffered from several types of errors. One source of these errors was from the LiDAR acquisitions themselves, which often did not entirely overlap or were collected improperly. Missing areas can often be observed between the gaps of the 49 LiDAR acquisitions over the region. In central Connecticut and eastern Maine, portions of the LiDAR were acquired with improper settings, resulting in forest vegetation being severely under-represented.

Banding errors occurred with forest attributes that had moderate to worse performance (tree count, percent conifer, and species/volume estimates). These bands were more likely to occur in areas where pulse density fell below 3 pls m$^{-2}$ and followed scan angle trends. In environments with a low pulse density and high scan angle, these attributes were often underestimated, possibly owing to less horizontal structure being captured by the LiDAR. Similar studies have normalized voxelized point clouds by applying a Beer–Lambert transformation to columns of voxels, and such an approach may have helped to normalize pulse density differences and possibly reduce banding [86]. Maps estimating tree size, such as biomass, volume, and basal area, had few banding errors.

### 4.4. Our Results in Context

Several previous studies have mapped aboveground biomass in this region and can be used to place the CNN model's performance in context. In one example, Kellndorfer et al. (2013) used Landsat and radar to map biomass across the continental United States (CONUS) and achieved RMSE values ranging from 42 to 48 Mg ha$^{-1}$ over New England with 30 m pixels. Qualitatively, these maps appear overgeneralized in comparison to our own. In 2008, Blackard et al. [87] used MODIS to predict biomass across the CONUS at a 250 m resolution and achieved an average absolute error in New England ranging from 49.7 to 60.4 Mg ha$^{-1}$. In a more regional study, Cartus et al. [88] mapped aboveground biomass in the Northeastern United States using radar and achieved RMSE estimates from 46 to 47.3 Mg ha$^{-1}$ with 150 m pixels, but noted that increasing pixel size dramatically reduced error. A more recent study mapped biomass in New England and Atlantic Canada using Landsat time-series data and achieved an RMSE of 44.7 Mg ha$^{-1}$ using 30 m pixels [57]. In the context of these studies, our aboveground biomass error of 36.9 Mg ha$^{-1}$ at a roughly 20 m resolution (FIA plot-level error) represents a considerable improvement over existing remotely-sensed regional estimates.

Localized studies in experimental forests throughout the region can also be used for comparison. Hayashi et al. [10] mapped stem volume using LiDAR and achieved RMSEs of 46 m$^3$ ha$^{-1}$ and 82 m$^3$ ha$^{-1}$ in two experimental forests in Maine and New Brunswick (the CNN achieved an error of 62.5 m$^3$ ha$^{-1}$). In a similar study, Hayashi et al. (2014) obtained RMSEs of 4993 trees ha$^{-1}$, 3.68 cm for QMD, and 13 m$^2$ ha$^{-1}$ for basal area, using 20 m cells at an experimental forest in Maine [70]. Our regional models achieved errors of 300 trees ha$^{-1}$, 5.2 cm QMD, and 7.9 m$^2$ ha$^{-1}$ basal area, thereby outperforming the local models in estimating tree count and basal area. Another study at an experimental forest in Massachusetts used large footprint LiDAR and radar to estimate biomass, achieving a RMSE of 66.6 Mg ha$^{-1}$ with 25 m cells [89]. Taken collectively, these results suggest that our regional model performs on par or better than local modeling efforts to predict the same forest attributes.

*4.5. Conclusions*

In this study, we mapped the forests of New England at a 10 m resolution, making estimates of 14 forest inventory attributes using harmonized LiDAR, satellite, and ground-based data. This was achieved through the use of disparate LiDAR datasets as well as satellite spectral, phenological, and disturbance data. Our method of modeling these attributes was somewhat novel and made use of deep learning by employing three-dimensional convolutional neural networks to scan the LiDAR point clouds, which are a form of deep learning. The CNN deep learning model outperformed traditional modeling approaches in most situations and proved useful for large-scale mapping, making use of disparate data and increasing data management and computational efficiency.

We validated the CNN-derived forest inventory maps using a region-wide external dataset derived from the USFS's FIA program. We concluded that the most successful estimates were of attributes that quantified tree size, moderately successful estimates were those that quantified tree density or percent coniferous, and less successful estimates were those that quantified more specific species groupings. In particular, we found our biomass estimates agree strongly with those of the FIA across the region.

We believe that both the deep learning models and the maps generated by this study will prove useful in further studies. In particular, the weights from the CNN model trained here could be used to initiate the training of models making estimates over different forested regions or for other LiDAR-related remote sensing problems. Likewise, the maps developed here can assist with wildlife habitat mapping, precision forestry, and carbon stock estimation in the region, as well as forming a large-scale baseline for future land-use change assessments and disturbance studies.

**Author Contributions:** Conceptualization, E.A., D.J.H., J.B.K., A.R.W., S.F.; methodology, E.A., D.J.H., J.B.K.; Software, E.A., J.B.K.; validation, E.A.; formal analysis, E.A.; investigation, E.A.; resources, D.J.H., A.R.W., B.D.C., J.A.K.J.; data curation, E.A., D.J.H.; writing—original draft preparation, E.A.; writing—review and editing, E.A., D.J.H., J.B.K., S.F., J.A.K.J., B.D.C. and A.R.W.; visualization, E.A.; supervision, D.J.H.; project administration, D.J.H.; funding Acquisition D.J.H. All authors have read and agreed to the published version of the manuscript.

**Funding:** This project was supported by the USDA National Institute of Food and Agriculture, Hatch (or McIntire-Stennis, Animal Health, etc.). Project number ME0-41907 through the Maine Agricultural & Forest Experiment Station and National Science Foundation RII Track-2 FEC (Award #1920908). Maine Agricultural and Forest Experiment Publication Number 3866.

**Institutional Review Board Statement:** Not applicable.

**Informed Consent Statement:** Not applicable.

**Data Availability Statement:** Mapped forest attributes can be accessed through the Environmental Data Initiative at the following URL: https://portal.edirepository.org/nis/mapbrowse?scope=edi&identifier=1007, accessed on 13 October 2021, doi:10.6073/pasta/e0dfb8220980bad60d1edb24ebd9d055.

**Acknowledgments:** We thank NASA Goddard's G-LiHT team and NEON for the use of their LiDAR. We thank Kathryn Miller and the National Park Service for the use of their Acadia National Park field data. We thank Eben Sypitowski and Baxter State Park for the use of their Scientific Forest Management Area CFI field plots. We thank Brian Roth and the University of Maine's Cooperative Forestry Research Unit for the use of their LiDAR and field data, as well as for considerable technical assistance. We thank Keith Kanoti and the University of Maine for the use of their CFI field plots. We thank the New Hampshire Division of Forests and Lands, Caroline A. Fox Research and Demonstration Forest for the use of their CFI field plot data. We thank David Orwig and Harvard Forest for the use of their Megaplot field inventory. We thank Noonan Research Forest for the use of their stem mapped plots. We thank Laura Kenefic and the U.S. Forest Service's Penobscot Experimental Forest for the use of their LiDAR and field inventory data.

**Conflicts of Interest:** The authors declare no conflict of interest.

## Appendix A

**Table A1.** A list of field inventories used for model training and the first phase of validation. In addition, the area those inventories represented, the number of LiDAR field plots, the number of LiDAR acquisitions, and acquisition characteristics are included. When inventories covered multiple sites, those sites are listed.

| Inventory | Sites | Location (s) | Area (km$^2$) | Number of Plots | LiDAR Acquisitions | Seasonality | Mean Pulse Densities (pls m$^{-2}$) | Temporal Field/LiDAR Discrepancy (Years) |
|---|---|---|---|---|---|---|---|---|
| Acadia National Park | Mount Desert<br>Isle au Haut<br>Schoodic Point | −68.294, 44.339<br>−68.627, 44.032<br>−68.065, 44.351 | 671 | 128 | 2 | Leaf on/off | 1.5, 12 | 0 to +2<br>−2 to +2 |
| Baxter State Park | Scientific Forest Management Area | −69.000, 46.176 | 87 | 882 | 1 | Leaf off | 5 | −3 |
| Bartlett Experimental Forest Echidna | | −71.286, 44.064 | 0.1 | 46 | 1 | Leaf on | 4 | −7 |
| Cooperative Forestry Research Unit | Austin Pond<br>Alder Stream<br>Dow Road<br>Golden Road<br>Harlow Road<br>Katahdin Ironworks<br>Lazy Tom<br>Lake Macwahoc<br>Penobscot Experimental Forest<br>Ronco Cove<br>Rump Road<br>Sarah Road<br>Schoolbus Road<br>St. Aurelie<br>Summit<br>Week's Brook<br>Weymouth Point | −69.705, 45.193<br>−69.798, 45.369<br>−69.609, 45.996<br>−68.675, 45.719<br>−67.842, 45.646<br>−69.367, 45.489<br>−69.456, 45.726<br>−68.286, 45.798<br>−68.608, 44.844<br>−69.634, 45.680<br>−71.018, 45.193<br>−70.911, 44.817<br>−70.778, 44.841<br>−70.161, 46.259<br>−68.480, 45.096<br>−68.522, 46.217<br>−69.308, 45.947 | 30,000 | 935 | 3 | Leaf on/off | 1.5,6, 12 | +2, +1, 0 |
| Carbon Monitoring System | | −69.764, 45.589 | 4645 | 414 | 3 | Leaf on/off | 8, 5, 15 | −2, −4, −5 |
| University of Maine Forests | Demeritt Forest<br>Penobscot Experimental Forest | −68.678, 44.933<br>−68.608, 44.844 | 55 | 912 | 3 | Leaf on/off | 1.5, 12, 6 | −8 to +2<br>−9 to +2<br>−10 to +2 |

**Table A1.** *Cont.*

| Inventory | Sites | Location (s) | Area (km$^2$) | Number of Plots | LiDAR Acquisitions | Seasonality | Mean Pulse Densities (pls m$^{-2}$) | Temporal Field/LiDAR Discrepancy (Years) |
|---|---|---|---|---|---|---|---|---|
| Fox Forest | | −71.911, 43.138 | 9 | 581 | 1 | Leaf off | 6 | −5 |
| Harvard Forest Megaplot | | −72.176, 42.538 | 0.4 | 6646 | 2 | Leaf on/off | 5, 12 | +1, +2 |
| Harvard Forest Echidna | | −72.182, 42.531 | 0.1 | 90 | 2 | Leaf on/off | 5, 12 | +6, +7 |
| Holt Experimental Forest | | −69.772, 43.871 | 0.1 | 1001 | 3 | Leaf on/off | 2, 12, 15 | −3, −5, −8 |
| Howland Experimental Forest | | −68.742, 45.206 | 2 | 556 | 2 | Leaf on/off | 5, 12 | +1, +2 |
| Howland Echidna | | −68.742, 45.206 | 0.1 | 80 | 2 | Leaf on/off | 5, 12 | −9, −5 |
| Noonan Research Forest | | −66.439, 45.977 | 0.1 | 25 | 1 | Leaf on | 5 | 0 |
| Penobscot Experimental Forest | | −68.608, 44.844 | 4 | 409 | 3 | Leaf on/off | 1.15, 12, 6 | −8 to +2 −10 to +2 −10 to +2 |
| Null Plots | | Regional | | 500 | 1 | Leaf on/off | 1.5 to 15 | 0 |

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
