# Peer review of "Synthesizing Disparate LiDAR and Satellite Datasets through Deep Learning to Generate Wall-to-Wall Regional Inventories for the Complex, Mixed-Species Forests of the Eastern United States"

_remotesensing, doi:10.3390/rs13245113_

Round 1

Reviewer 1 Report

Dear editor,

This work approaches the use of CNN in forest attribute predictions based on area-based approach. Models were trained using LiDAR and satellite data and assessed using three validation strategies: testing datasets, validation using external data, and validation based on probability-based countrywide inventory. Random Forest approach was used as the baseline for comparison. The results show that CNN is promising when it comes to above-ground biomass estimation, for instance, but not to other attributes such as stand densities.

The work is pertinent for publication. The text is in general well written and well presented, although it has some issues that must be clarified by the authors, so a major correction is recommended. The main issues are described below, and specific comments are pointed later.

GENERAL COMMENTS:

  1. In L308 the authors mentioned the problems regarding the FIA plots explaining why they did not use them for model training. However, these same FIA data were used for model validation. Although the authors tried to justify this in L297-307, it is still not clear how such validation would provide liable indications of the model accuracies. Please, check the work of Packalen et al. (2019) for your argumentation.

Packalen, P., Strunk, J., Packalen, T., Maltamo, M., & Mehtätalo, L. (2019). Resolution dependence in an area-based approach to forest inventory with airborne laser scanning. Remote sensing of environment, 224, 192-201. https://doi.org/10.1016/j.rse.2019.01.022

  1. L331. I did not understand this table 3. In L329 the authors mentioned that just the CNN with satellite metrics were used in this phase two validation, but Table 3 caption indicates three models, and they also mention Random Forest. Besides, there are no green highlights in this table…

  1. In L274 they mention that metrics in RF models were selected using the variable importance approach but forgot to mention how many metrics were used at the end. Besides, some works have already demonstrated that variable selection in RF is quite unnecessary, although it is not a problem at all (e.g., Cosenza, D. N., Korhonen, L., Maltamo, M., Packalen, P., Strunk, J. L., Næsset, E., ... & Tomé, M. (2021). Comparison of linear regression, k-nearest neighbour and random forest methods in airborne laser-scanning-based prediction of growing stock. Forestry, 94(2), 311-323. https://doi.org/10.1093/forestry/cpaa034).

  1. The authors could have briefly mentioned somewhere in the discussion or introduction about other machine learning approaches able to predict for multiple responses, such as kNN (e.g., Packalen, P., Temesgen, H., & Maltamo, M. (2012). Variable selection strategies for nearest neighbor imputation methods used in remote sensing based forest inventory. Canadian Journal of Remote Sensing, 38(5), 557-569. https://doi.org/10.5589/m12-046).

  1. Although the authors compared the CNN estimations with traditional forest inventories (i.e., probability-based inference), they did not consider the application of CNN for ALS-assisted inference. I think they should at least complement the discussion in this sense. For instance, CNN models are quite computationally intensive, so computing bootstrap estimators for model-based inference would be impractical, but it would might not be the case for model-assisted estimators. Two works might be helpful in this regard:

Gregoire, T. G., Næsset, E., McRoberts, R. E., Ståhl, G., Andersen, H. E., Gobakken, T., ... & Nelson, R. (2016). Statistical rigor in LiDAR-assisted estimation of aboveground forest biomass. Remote Sensing of Environment173, 98-108. https://doi.org/10.1016/j.rse.2015.11.012

McRoberts, R. E., Magnussen, S., Tomppo, E. O., & Chirici, G. (2011). Parametric, bootstrap, and jackknife variance estimators for the k-Nearest Neighbors technique with illustrations using forest inventory and satellite image data. Remote Sensing of Environment115(12), 3165-3174. https://doi.org/10.1016/j.rse.2011.07.002

SPECIFIC COMMENTS:

L230-231. Please put the reference in the journal standard as possible.

L288. Is there any reference informing the results of these county-level estimates, or have you calculated the ground-truth inventory by yourselves?

L325. Confusing sentence (“In terms of RMSE, the CNN with satellite
metrics outperformed the one without 100 % of the time.”). Please rephrase it

L324. Maybe it is better to remove the term “significantly” in this sentence because you have not conducted any statistical test in this sense—maybe you could replace it by “consistently”. Besides, it makes the paragraph rather contradictory when we read in L327 that the satellite data were not so important in proportional terms. Actually, this is  already expected (e.g., Fassnacht, F. E., Hartig, F., Latifi, H., Berger, C., Hernández, J., Corvalán, P., & Koch, B. (2014). Importance of sample size, data type and prediction method for remote sensing-based estimations of aboveground forest biomass. Remote Sensing of Environment154, 102-114. https://doi.org/10.1016/j.rse.2014.07.028)

L326. Correct to: “…in predicting 3 out of 14 (21.4 %) attributes.”

L332. Please, provide the RMSE and bias formulas (they might be in the Table 3 footnote)

L335. I think that justifying that RF models were not used here due to computational effort is unnecessary since you have already mentioned that the second phase of validation would be conducted with the winning model of the first phase (L284). Besides, I presume that CNN models would be much more computationally intensive to be applied than RF models.

L340-340. I do not comprehend this statement. It is better you contextualize it first and give it at the end of the paragraph. But more important, how can you ensure that the increased two-phase validation error is not due to the differences in the field plots? (see also the general comment 1) Maybe the representativeness of the phase one validation data could have been checked using histograms and other exploratory analyses. Have you conducted checked this in your previous analysis?

L404. Please change the colors used in this table (yellow is almost invisible, for instance). Consider using symbols instead, like *, **, ***…

L462. Produced or yielded?

Reviewer 2 Report

This manuscript presents a timely and innovative fusion of lidar and additional remote sensing data with FIA/Inventory data to broadly map forest attributes of keen research and industrial interest. I was very excited to read the abstract in the review request and promptly reviewed this paper within the next couple of hours. And overall this is a really great manuscript. I especially like the detailed descriptions in the method, the included data set generated, and the attention to detail regarding the theory and application of all of the methods used. I only have minor comments which follow. Thank you for this opportunity and I look forward to the final version of this manuscript. 

Comments:

Line 57 – I would avoid using the word “unfortunately” here as there is a strong case to be made for additional information gain regardless of collinearity. Perhaps caveat with something to the effect “this may or may not be optimal depending on use case”

Line 60-71 – Is the pulse density concern a low threshold issue? That after some sufficient threshold that pulse density becomes less of an issue?

- This is brought up late in section 1.2, but there is an issue with comparisons between leaf-on and leaf-off that are not well addressed explicitly. I am thinking groups like NEON who are making leaf-on acquisitions and then statewide programs in the US like in Pennsylvania, Indiana, NC where leaf-off data are also being used for forest inventory/analysis. These apples to oranges nature of this is problematic. You could more explicitly detail this perhaps. I see the reference to ”phenology” earlier now upon re-read.

Line 75-78 –  “….aforementioned obstacles” of regression and RF?  Ca

Section 1.3 – I am really into what you all are doing here. I don’t see this paper that from Citui et al. 2017 (https://doi.org/10.1111/2041-210X.12921) cited, but it is very inline with the idea of trying to use “all” the data via a PCA approach and should probably be cited in here. Apologies if it is and I don’t see or can’t ctrl-f it. MDPI citation schemes are the worst honestly.

Section 1.3  - verb tense is a little all over the place through here as is some passive voice.

Line 136 – how large is large?

Line 140 – reference needed

Line 152 – This approach is so good I think.

Section 2.3.1 – This point I am interested to learn more on-the leafon and leaf off model development. Maybe you can clarify that more? Maybe that comes up later. I will keep reading.

Line 214 – which indices of greenness and senescence did you use? Thresholds etc. Can you clarify how this was defined?

Section 2.4.1 – did you use raw lidar return density or were there calculations for occlusion? Beer-Lambert or Mac-Horn ?

Figure 2. – Can you make this full page?

Line 248 – Can you explain a bit more about how equation 1 is used?

Section 2.4.3 – This section brings up the practical use of this approach. Who is going to be able to do this method? Is it practical at large scales?

Line 264 – yes, thank you. The intensity metrics are troublesome.

Line 271 – “Although random forest is nontraditional in that it is non-parametric, we refer to it as such because it has become widespread in the field of LiDAR modeling, and has been a recommended modeling technique for many years”   This sentence is confusing as heck. Are you calling RF parametric? Traditional?

Lines 291 – Some kind of citation for FIA plot data is needed. Additionally, did you use the randomized data or the “real” data?   Ah! I see in Figure 3 caption that these are the fuzzed plot locations. Please address and talk about the location error with that more explicitly. This is funded by USDA right? Can you get access to unfuzzed data? I realize it can be a bear.

Figure 3. If the warmer colors mean greater observations, then why is there a different scale and variable for each map? Please clarify.

Figure 4. This is a very thought provoking plot. I like it.

Figure 5. The caption mentions “red areas” but I  don’t see any. IS something missing?

Discussion – The outperformance of RF models is an interesting finding, back to a previous question I raised, how scalable is the use of this approach though? Do I need high performance computing access? GPU based processing?

Figures  6-8 are great, but would be better in Viridis (like fig 4) to account for color blindness. Great plots though otherwise.

Round 2

Reviewer 1 Report

Dear editor,

This is the review I could do in this dead line of three days.

My comments were properly answered by the authors. I recommend for acceptance, but the authors must care about the text length. The text has paragraphs and phases that could be removed (or shrunk) to improve the reading flow. I believe this could done the introduction and discussion, for instance, in L7-82, 84-97, 568-579.

Other minor issues are described below.

L199. Here you mentioned about “each model” but the reader might not understand that you are actually referring to CNN and RF models, while RF approach was just briefly mentioned in the introduction).  

L200. The majority of LiDAR data?

L250. There are still some references out of the journal standard (L250, L251, 596, 611…)

L283. Please, provide the complete reference of the rLiDAR package in the references. You can use the R function citation(“rLiDAR”) to help you